# Individual, Family, and Social Factors Associated with Gestational Weight Gain in Adolescents: A Scoping Review

**DOI:** 10.3390/nu15061530

**Published:** 2023-03-22

**Authors:** Reyna Sámano, Hugo Martínez-Rojano, Luis Ortiz-Hernández, Oralia Nájera-Medina, Gabriela Chico-Barba, Ricardo Gamboa, María Eugenia Mendoza-Flores

**Affiliations:** 1Programa de Posgrado Doctorado en Ciencias Biológicas y de la Salud, División de Ciencias Biológicas y de la Salud, Universidad Autónoma Metropolitana, Mexico City 04960, Mexico; lortiz@correo.xoc.uam.mx (L.O.-H.); onajera@correo.xoc.uam.mx (O.N.-M.); 2Coordinación de Nutrición y Bioprogramación, Instituto Nacional de Perinatología, Secretaría de Salud, Mexico City 11000, Mexico; gabyc3@gmail.com (G.C.-B.);; 3Sección de Posgrado e Investigación de la Escuela Superior de Medicina del Instituto Politécnico Nacional, Mexico City 11340, Mexico; 4Departamento de Atención a la Salud, Ciencias Biológicas y de la Salud, Universidad Autónoma Metropolitana-Xochimilco, Mexico City 04960, Mexico; 5Programa de Maestría y Doctorado en Ciencias Médicas, Odontológicas y de la Salud, Universidad Nacional Autónoma de México, Mexico City 04510, Mexico; 6Departamento de Fisiología, Instituto Nacional de Cardiología, Mexico City 14080, Mexico; rgamboaa_2000@yahoo.com

**Keywords:** gestational weight gain, teen, pre-pregnancy body mass index, family, age, pregnancy in adolescence

## Abstract

About 56% to 84% of pregnant adolescents have inappropriate (insufficient or excessive) gestational weight gain (GWG); however, the factors associated with GWG in this age group have not been systematically identified. This scoping review aimed to synthesize the available scientific evidence on the association of individual, family, and social factors with inappropriate gestational weight gain in pregnant adolescents. To carry out this review, the MEDLINE, Scopus, Web of Science, and Google Scholar databases were searched for articles from recent years. The evidence was organized according to individual, family, and social factors. The analyzed studies included 1571 adolescents from six retrospective cohorts, 568 from three prospective cohorts, 165 from a case–control study, 395 from a cross-sectional study, and 78,001 from two national representative samples in the USA. At the individual level, in approximately half of the studies, the pre-pregnancy body mass index (pBMI) was positively associated with the GWG recommended by the Institute of Medicine of the USA (IOM). The evidence was insufficient for the other factors (maternal age, number of deliveries, and family support) to determine an association. According to the review, we concluded that pBMI was positively associated with the GWG. More quality studies are needed to assess the association between GWG and individual, family, and social factors.

## 1. Introduction

Teenage pregnancies in developing countries account for over 90% of all reported cases worldwide [1]. Adolescent pregnancy is a high-risk condition associated with a higher probability of adverse maternal and neonatal outcomes, such as delayed intrauterine growth and development and low birth weight [2,3]. Pregnant adolescents are a vulnerable group since their longitudinal growth is compromised [4,5,6] and, generally, they consume an excessive or deficient diet in terms of quantity and/or quality [7,8,9], in addition to being exposed to adverse psychosocial factors that may influence gestational weight gain [10,11]. As a result, pregnant adolescents worldwide have inappropriate rates of gestational weight gain (GWG). Between 56% and 84% of adolescents typically have inappropriate GWG, either insufficient or excessive [12,13,14,15,16]. Pregnant adolescents with pre-pregnancy overweight and obesity have the highest prevalence of excessive GWG, so a better understanding of the factors associated with inappropriate GWG is a public health priority.

The risks of excess GWG include, but are not limited to, postpartum weight retention for the mother and the birth of a macrosomic or large (for the gestational age) newborn and subsequent childhood overweight or obesity [17], as well as metabolic programming for chronic non-communicable diseases in offspring [12,13]. Furthermore, excess GWG may contribute to overweight and obesity in postpartum women as well as their children, with obesity perpetuating for several generations [13,14]. Excess GWG can contribute to postpartum weight retention; when this weight gain occurs during adolescence, this weight can remain and even increase up to 18 years after the first pregnancy [14]. This fact is relevant, considering that the prevalence of overweight and obesity before pregnancy among adolescents [15,16] has increased in low- and middle-income countries, reaching up to 45% [17,18,19,20].

On the other hand, the conceptual framework of the Institute of Medicine of the United States of America (IOM), which is a non-governmental organization made up of various committees of experts for studying GWG and various health issues, has established guidelines for gestational weight gain that include sociocultural, environmental, and maternal factors, many of which have been studied as potential factors associated with insufficient or excessive gestational weight gain in pregnant adults [4,21,22,23]. However, regarding pregnant adolescents, it is unknown if these factors have been studied or if they are associated with inappropriate gestational weight gain. Despite the magnitude of the problem, we could not identify any previously published reviews that systematically identified the factors associated with GWG in pregnant adolescents. Consequently, this review aimed to present up-to-date evidence on this topic and reveal understudied areas that need further investigation. The purpose was to synthesize the available scientific evidence on the association of individual, family, and sociocultural factors with inappropriate GWG in pregnant adolescents.

## 2. Material and Methods

### 2.1. Design

We performed a scoping review using the Preferred Reporting Items for Systematic Reviews and Meta-Analyses (PRISMA) guidelines [24]. Our review did not imply any risk because it consisted of a review of previously performed research previously that complied with ethical aspects. Moreover, the present review was carried out in line with PRISMA [24] guidelines and was derived from a protocol approved by the research and ethical institutional committees at the Instituto Nacional de Perinatología in Mexico City (2017-2-101 INPer CONACyT FOSSIS SALUD-2018-01-A3-S-40575).

### 2.2. Inclusion Criteria and Selected Studies

Eligible studies were original articles with an observational design published since 1990, when GWG in pregnant women was considered an event of interest. We included studies that defined the GWG as the increase in body weight during pregnancy in kilograms or pounds and those studies that assessed the GWG in categories [21,25] of women between 10 and 19 years old. We included studies with adult women as long as they showed the results separated by age group (for example, stratified analyses). We included articles written in English, Portuguese, and Spanish. We excluded animal and in vitro models, publications that came out before the results were published, case studies, series of cases, reviews, book chapters, letters to the editor, editorials, opinion articles, comments or errata, and summaries. Studies published only as abstracts were excluded because their quality could not be adequately assessed, as were duplicate or secondary publications on the same risk exposure and population to avoid multiple publication bias. Studies were also excluded if they included adolescents with previous chronic diseases such as cancer, systemic lupus erythematosus, rheumatoid arthritis, heart disease, or an endocrine disease.

### 2.3. Data Sources

The following databases were consulted: MEDLINE, Scopus, Web of Science, and Google Scholar. MeSH (Medical Subject Headings) terms and DeCS (Descriptors in Health Sciences) were used for three categories: individual, family, and sociocultural factors. A search framework was applied for extensive research with a wide scope to identify the applicable studies. There were different search strategies used for each database (see Appendix A).

Figure 1 shows the selection process. We identified 842 titles in a search of the combined databases. We eliminated 298 duplicated titles. For the remaining 554 titles, we selected the articles based on their title and summary. Finally, there were 13 original articles.

### 2.4. Organization of the Information

We summarized the findings and organized them according to the guidelines for scoping reviews [26,27]. We identified the design, quality, temporality, sample size, country, and other individual variables in them. GWG was presented as a percentage or in kilograms, according to the source. Topics were organized according to the associated factors: individual, family, and sociocultural.

### 2.5. Assessment of the Quality

Each research was evaluated independently by two authors (R.S. and H.M.-R.) independently. They reviewed the titles, backgrounds, and all articles using predefined criteria [28,29]. The quality of the cross-sectional and longitudinal studies was examined by applying 14 questions with two answers with the following scores: yes, 1; no, 0; not applicable, 1; uninformative, 0; could not be determined, 1. The total score was used to classify the quality of the research according to the guidelines of the assessment tool of quality of the National Institutes of Health (NIH). The highest score was 14, with 12–14 points classified as high quality, 9–11 as good quality, 7–8 as fair, and <7 as poor [30,31]. The agreement between the two reviewers for the whole text was evaluated by Cohen’s weighted kappa statistic. Later, three authors (L.O.-H., O.N.-M., and G.C.-B.) analyzed the information from the articles. They confirmed the quality of the research; they curated and corrected any content (see Appendix A).

## 3. Results

### 3.1. General Data

From 842 titles, 13 were included in the review. The value of Cohen’s weighted kappa (k) for the initial agreement between the two authors for all articles combined was 0.924.

The samples from all the articles analyzed were 1571 adolescents from six retrospective cohorts [32,33,34,35,36,37], 568 from three prospective cohorts [38,39,40], 165 from a case–control study [41], and 395 from a cross-sectional study [42]. In addition, we included two studies with nationally representative samples of 78,001 adolescent women from the USA [9,43]. The rest of the reviews considered community and hospital samples.

Over the last 14 years, the USA has been the country with the most studies on pregnant adolescents and their association with GWG [9,32,33,34,35,36,38,43], followed by studies performed in Mexico [40,41,42], and, finally, studies in other countries such as Turkey [37] and Colombia [39]. Furthermore, most of the studies were performed in low-income countries [32,35,36,38,39,40,41,42], without government support for medical services or health insurance [32,35,39,40,41,42], and in vulnerable urban or suburban areas [32,35,36,38,39,40,41,42], with a variety in the levels of the studies’ quality.

According to the evidence, the frequency of excessive GWG among adolescents was between 38% and 50%. At the same time, the frequency of inadequate GWG was 18% to 34%. Therefore, 56–84% of adolescents had inappropriate GWG, and only one-third of the participants had adequate GWG.

### 3.2. Individual Factors

We identified six studies [9,32,36,38,40,43] that described that adolescents with a higher pregestational body mass index (pBMI) had a higher frequency of excessive GWG (Table 1). In contrast, three studies did not find an association between pBMI and GWG [33,35,37]. Although those studies did not find any association with the GWG as reported in kilograms or pounds, they did not specify if the GWG was according to the IOM’s weight gain recommendations or not [33,35]. Both studies were of good quality.

The association between age and GWG was analyzed in 330 pregnant African American adolescents [34] (Table 1). This study did not show any association between age and GWG.

Regarding the number of deliveries, among adolescents from the USA, the number of deliveries was associated with GWG, with a higher number of deliveries associated with lower GWG [43] (Table 1). Nevertheless, Timur et al. [37] did not report any difference in pBMI between the first (23, range 19–35) and the second pregnancy (25, range 20–37; *p* = 0.672) or in GWG in the first (12.4 ± 5 kg) and second pregnancies (11.5 ± 6; *p* = 0.462). Although there are inconsistencies, it seems that the first pregnancy can contribute to postpartum retention in adolescents [18].

Other individual factors in our analysis included nutritional intake, mental health, lifestyle, and other biological factors. For example, a study with a sample of Mexican pregnant adolescents reported a negative correlation between GWG and the adequacy of energy intake as a percentage (r = −0.227, *p* = 0.003) (Table 1) [40]. Nevertheless, we did not identify research on the development of negative psychological states (such as depression, anxiety, and stress) or physical activity and their association with GWG in adolescents. Finally, in two studies from Latin America (Mexico and Colombia), serum leptin [40], insulin, adiponectin levels, and a homeostatic model assessment for insulin resistance (HOMA-IR) predicted the variation in excessive GWG [39] or were associated with higher pBMI (see Table 1).

### 3.3. Family Factors

In a sample of Mexican adolescents [42], it was reported that maternal and neonatal outcomes did not differ according to the size of the family support network or whether the principal member of the family support network was the adolescent’s mother. Additionally, this research demonstrated a higher probability of having a newborn who was small for their gestational age when the first members of the network were non-blood relatives of the pregnant adolescents (e.g., mother-in-law) (see Table 2).

### 3.4. Sociocultural Factors

A sample of pregnant Mexican adolescents with a history of sexual abuse presented with a 5 kg lower GWG than those without sexual abuse. The offspring’s birth weight and length were higher in adolescents without a history of sexual abuse [41]. Race or ethnicity was mentioned in three studies [32,33,36]. However, only one [32] reported that Latina adolescents had a higher frequency of inappropriate GWG than the other groups. We did not find articles that addressed sociocultural factors and their association with GWG in pregnant adolescents (see Table 2).

## 4. Discussion

Identifying and analyzing the factors associated with GWG in pregnant adolescents is relevant because many have excessive and inadequate GWG. In addition, the GWG is a modifiable factor; it is known to affect maternal and fetal outcomes such as birth weight and adiposity. The present review is one of the first to describe the association between GWG and individual, family, and sociocultural factors.

### 4.1. Findings

pBMI is the individual factor that has been analyzed most frequently. Some research where GWG was analyzed according to guidelines from the IOM as a categorical variable reported an association with pBMI. Nevertheless, those that reported GWG as a continuous variable did not report any association with pBMI [33,35]. However, they did not consider other factors as possible reasons for the disparity/heterogeneity in their results, such as income and ethnicity, country of origin, design (prospective/retrospective), and the pBMI criteria.

Figure 2 summarizes the factors associated with GWG in pregnant adolescents, as well as the factors that may possibly be associated.

Regarding other variables, we did not identify any associations because of the low quantity and quality of the scientific research. Hence, these articles were not included in our review.

### 4.2. Individual Factors

Our scoping review showed that nearly half of the relevant research on pregnant adolescents found that pBMI was positively associated with GWG [9,32,36,38,40]. Adolescents who became pregnant with a high pBMI presented a higher frequency of excessive GWG. The fact that they began pregnancy with a high pBMI increases the probability of obesity in early adulthood [44,45,46,47].

When GWG was reported in kilograms or pounds, total GWG was often similar in all categories of pBMI. Nevertheless, we should consider that the recommendations of the IOM regarding pBMI are more restricted for overweight and obesity. In that case, adolescents with pregestational obesity could have a high frequency of excessive GWG [9,33,35,38]. Thus, they could have misreported their GWG because overweight and obese adolescents tend to gain the same weight as normal or low-weight adolescents [33,38,48]. Therefore, assessments of GWG without controlling for pBMI categories did not allow the identification of an adequate GWG, especially among pregnant adolescents who were overweight or obese. Hence, we should be cautious about generating any conclusions about GWG and pBMI in the case of GWG as a continuous variable.

In addition, only one investigation that evaluated the adolescents’ age at the time of pregnancy and their GWG was included in this review; in this study, no association between the adolescents’ age and GWG was reported [34]. Similar to what has been reported in other studies where adolescents and pregnant adults were compared, gestational weight gain was similar in both groups [9,11,48,49,50,51,52]. In contrast, there have been studies where it was shown that pregnant adolescents gained more weight compared with pregnant adults [43,44,45,53]. The latter may be because some adolescents began their pregnancy overweight, obese, or with a lower BMI and, therefore, were expected to gain more weight to compensate for the needs of the pregnancy itself. This is because when the recommended weight gain for pBMI was assessed, it was in line with the US Institute of Medicine recommendations for adults [47].

Regarding the association between the number of deliveries and the GWG, it was shown in a study carried out on a group of adolescents that the fewer the number of deliveries, the greater the gestational weight gain [45]. Similarly, the frequency of excessive GWG in primiparous adolescents was shown to be higher than in their peers with multiple pregnancies [45]. Contrary to what was mentioned above, in another study with pregnant adolescents from Turkey, it was shown that there were no significant differences in pBMI and GWG between the first and second pregnancies [46]. In contrast to these findings, Groth et al. [14] demonstrated that gestational weight gain had long-term effects on the body mass index in African adolescents. Excessive weight gain during pregnancy is likely to contribute to long-term weight retention, especially if adolescent girls are overweight or obese when they become pregnant with their first child. Likewise, in pregnant adults, Hill et al. [54] performed a systematic review with a meta-analysis, where the main findings indicated that the number of children or pregnancies was positively associated with pBMI. In contrast, the association between the number of deliveries and the GWG was less clear in pregnant adolescents. The number of children was not directly associated with postpartum weight retention, so the association between the number of deliveries and GWG as well as postpartum weight retention remains unclear, and its influence is likely to be indirect and complex.

Consequently, the weight gain that occurs during the first pregnancy, especially if it is excessive and is not lost after delivery, influences the results of future pregnancies. In this regard, it is important to address both adult and adolescent nulliparous women for advice on how to avoid excess GWG [55]. Therefore, more research on this topic is currently warranted in order to understand the association between the number of deliveries and maternal obesity.

In the present analysis, we only found one study of Mexican adolescents describing the association of energy intake and serum leptin levels with GWG [40]. Therefore, we do not have enough evidence to establish any conclusion. Although no more studies focused on pregnant adolescents were found, in pregnant adults, it was reported that regardless of pBMI, the design, dietary assessment methods, and country, the changes in energy intake were not correlated with GWG [56]. This limited evidence should be considered with caution for pregnant adolescents. This information does not mean that the association between GWG and energy intake does not exist because of the inconsistency among studies on adolescents. Pregnant adolescents are a vulnerable group in terms of obesity or low weight. They can have adverse perinatal outcomes, probably caused by a poor diet [57]. Probably, this is due to a lack of dietary guidelines for pregnant adolescents with different pBMI levels. Health professionals must consider that pregnant adolescent have higher nutrient requirements than adults because adolescents have not yet finished their linear growth [4,5,58].

Scientific evidence of the role that hormones play in gestational weight gain in pregnant adolescents is still scarce [48,49]. So far, two investigations have been described. The first was in a small group of pregnant adolescents from Colombia, where it was shown that the concentrations of leptin, insulin, and the HOMA-IR index were positively correlated with BMI (r = 0.83 and *p* < 0.0001; r = 0.56 and *p* ≤ 0.0001, and r = 0.54 and *p* ≤ 0.0001, respectively) [39]. The second study was conducted in a group of Mexican adolescents, where it was shown that pregnant adolescents with leptin concentrations higher than 20 ng/mL had a higher GWG compared with adolescents with concentrations of <20 ng/mL [40]. As it can be seen, there are several metabolic pathways that regulate body weight control during pregnancy, and hormones, cytokines, and nutrients can be used as neurotransmitters, substrates, and/or catalysts. We can conclude that more research is needed on the role played by hormones, cytokines, macronutrients, and micronutrients in the GWG of pregnant adolescents, especially as the prevalence of overweight and obesity has increased considerably in the adolescent population in both developed and developing countries, such as Mexico, Chile, Canada, Turkey, Colombia, the USA, etc., [13,14]. The number of studies and participants could be higher and of better quality if there were more investigations with representative samples from different regions and ethnicities. Only two studies [39,40] offered a new approach to explain the probable association of hormones and GWG in pregnant adolescents.

Just one study [32] explained that pregnant adolescents with obesity or overweight who had experienced excessive GWG had a greater frequency of presenting with depressive symptoms in the postpartum period, which is a concern because it could lead to a vicious circle of obesity, depression, and anxiety. In addition, a sample of African American adolescents found that severe depression can be associated with excessive GWG [59]. However, other studies on women of all ages [60,61] did not demonstrate an association between depression and GWG [48,62].

However, our search did not find other studies that associated emotional health and GWG in pregnant adolescents. Even so, emotional health can indirectly be associated with GWG. Biological, psychological, and social mechanisms could be the basis of pBMI, depressive symptoms, and GWG [32,61]. For example, eating high-energy foods is a form of compensation for the negative affective or psychological states of some women. These behaviors can result in excessive energy intake and an accumulation of fat mass tissue during pregnancy [62,63]. Therefore, the lack of studies limits us from concluding that there is any association of GWG with pregnant adolescents’ emotional or psychological health.

### 4.3. Family and Sociocultural Factors

Regarding family factors, we identified one study [42] reporting that a lack of parental support was related to inadequate GWG. The lack of social or family support could negatively affect nutrient intake, antenatal care [64], and GWG. This effect is relevant when there are unfavorable economic, social, and emotional conditions [65], as is the case for most pregnant adolescents. Family support becomes fundamental, as adolescents have a high risk of inadequate or excessive GWG and birthing low birth-weight babies, as well as not meeting their needs as adolescents [7]. Nevertheless, there is insufficient evidence to draw any conclusion.

In our review, the number of articles on the sociocultural variables associated with GWG in pregnant adolescents was limited. Overall, adolescent mothers reside in rural or suburban areas. These regions are characterized by social insecurity, where child marriages, physical and sexual abuse, and inadequate GWG are common [41,66].

Most pregnant adolescents are economically vulnerable and come from poor and developing countries. Some pregnant adolescents analyzed in our scoping review were from the USA [66,67,68], a developed country, but their participants were those with a vulnerable economic status, similar to those in other countries [67,68,69,70]. Adolescents can have limited access and availability to free governmental health and reproductive services due to a lack of employment or the economic stability of their parents and families [67,68]. This context can cause inappropriate antenatal and nutritional care [71]. Therefore, it could affect their GWG.

Another potential factor associated with GWG is sociocultural. For example, a sample of Mexican adolescents frequently satisfied their food cravings [72], which could contribute to inappropriate GWG because they tended to eat for two, despite being aware that they should not eat like this. Furthermore, among different groups of pregnant Latina women with a low income living in the USA [73], eating more than the recommendations to satisfy their food cravings was culturally acceptable [74]. Therefore, these conditions could promote excessive GWG [75]. However, we needed more scientific evidence to find their association with GWG in adolescents.

The present review is one of the first on GWG in pregnant adolescents. Nevertheless, there is a lack of investigations focused exclusively on pregnant adolescents, particularly on the individual, family, and social factors. The precise role of the individual, family, and social factors on GWG in pregnant adolescents could strengthen the development of programs or policies to improve antenatal care. An example of this could be the “Nurse–Family Partnership”. This is a program of antenatal home visits by nurses to primiparous women from low-income settings. It has demonstrated its effectiveness in different parts of the world [76].

In light of our scoping review, we propose that future efforts be aligned along three axes. (1) Research should continue establishing the role of hormones and cytokines in adiposity. (2) It is necessary to explain the relationship between GWG and genetic variants in pregnant adolescents through new scientific studies. (3) Another avenue could be research on GWG, physical activity, and nutrient intake in pregnant adolescents.

We did not identify any research that analyzed the association between GWG and genetic variants in pregnant adolescents, although 20% of excessive GWG could be explained by genetic variability [77]. In this context, polymorphisms in the *LEPR* and *FTO* genes have been associated with GWG, at least in adult women, particularly in those beginning pregnancy with overweight or obesity [12,78]. Furthermore, the intake of rich lipid foods is associated with the *MC4R rs17782313* polymorphism [79] and the expression levels of the *FTO* gene [80]. Hence, epigenetic effects during pregnancy could contribute to helping to understand the mechanisms of GWG and pBMI in the long-term health outcomes of the mother and her offspring.

We also did not find any studies on the association between GWG and physical activity in pregnant adolescents. Therefore, this is another topic that we recommend addressing. Finally, it is recommended to highlight the valid tools that allow us to compare different studies on food and negative affective status.

In this present research, we documented that in nearly half of the studies, pBMI was positively related to GWG. This association indicates that health personnel should assess the pBMI of pregnant adolescents to determine the precise GWG and give adequate antenatal care, particularly to those with pre-pregnancy overweight or obesity [81,82]. Adolescent women, especially those under 15 years old, should use specific charts for BMI for age and gender, such as the growth charts of the WHO [83] or CDC [84], because pregnant adolescents could receive inappropriate recommendations on GWG according to IOM guidelines [82].

### 4.4. Strengths and Limitations

Our scoping review analyzed the association between GWG and individual, family, and social factors in pregnant adolescents. Even though our study had limitations, we did not search for scientific evidence in other databases. We searched Medline, Scopus, Web of Science, and Google Scholar, which are the central databases of health sciences. Another limitation was that the results from several disciplines dealing with pregnant adolescents could be published in non-indexed scientific journals, so it was possible that we lost some articles during our search process.

Because of the small sample size, other investigations still need to be carried out so that we can draw a conclusion from the findings. For example, another area for improvement was that one-third of the included studies were performed in developing countries with medium or poor methodological rigor and a variety of methods applied.

The principal strength of our scoping review is that we reported a gap in the research in the scientific literature on GWG among pregnant adolescents. At the same time, our scoping review established that the topic should continue to be studied to determine the specific role of macronutrients, hormones, cytokines, genetic variants, and physical activity, which could condition the increase in adiposity and promote excessive GWG among pregnant adolescents.

## 5. Conclusions

At the individual level, pBMI was positively associated with GWG as long as it was assessed according to IOM guidelines. Moreover, excessive GWG could probably play a role in postpartum weight retention in the long term, especially if adolescents began their first pregnancy with overweight or obesity. At the same time, family and social factors do not have enough scientific evidence to describe any association with GWG.

GWG in pregnant adolescents is still a complex phenomenon and remains critical in antenatal management. Future research is needed to fill the gap we reported in this review, focusing on GWG as an associated factor of chronic diseases and as a consequence of pBMI, especially in pregnant adolescents, whose growth and development can be limited by pregnancy.

## Figures and Tables

**Figure 1 nutrients-15-01530-f001:**
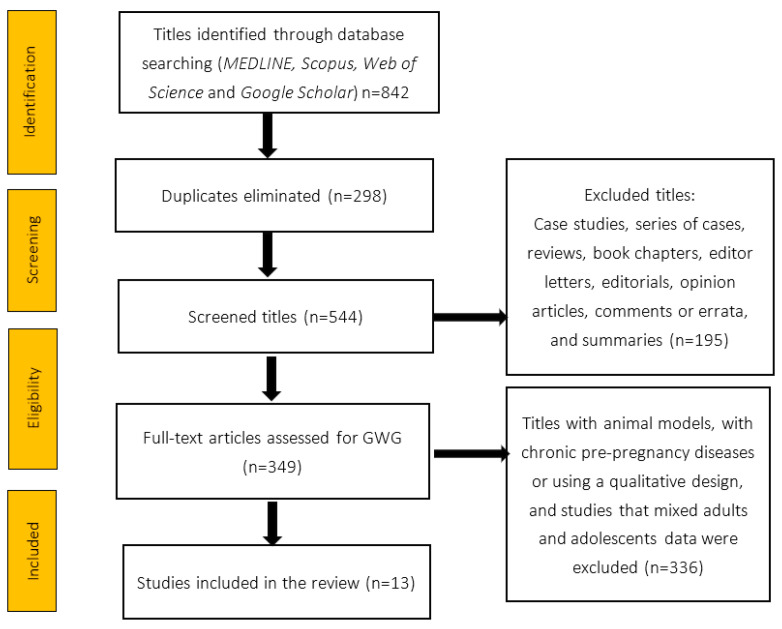
PRISMA flow diagram used for selecting the studies.

**Figure 2 nutrients-15-01530-f002:**
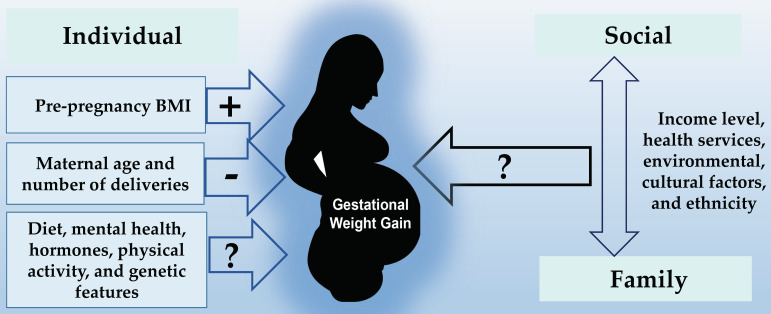
Factors associated with gestational weight gain in pregnant adolescents. Association: (+) positive, (−): negative, (?): uncertain.

**Table 1 nutrients-15-01530-t001:** Synthesis of studies on individual characteristics and gestational weight gain in pregnant adolescents.

Author, Year, Study Design	Participants	Exposure	GWG	Findings	Quality
Pre-pregnancy Body Mass Index (pBMI)					
Elchert et al. [9]. 2015 Retrospective cohort, 2006–2012	*n* = 326,368. Pregnant women stratified by maternal age: 0.3%, <15 years; 7.0%, 15–17 years; 14.8%, 18–19 years; 77.9%, 20–34 years. Representative sample United States	Distribution of pBMI (in %) <15 years old: 23.2 15–17 years old: 23.5 18–19 years old: 24.5 WHO age- and sex-specific BMI	Excessive GWG, in % <15 years: 59.8 15–17 years: 59.9 18–19 years: 62.6 IOM definition, 2009	pBMI was associated positively with GWG A large proportion of pregnancies had excessive GWG. The teens least likely to gain an adequate amount of weight were those who had a pBMI indicating obesity.	High, 12
Cunningham et al. [32]. 2018 Retrospective cohort, 2008–2012	*n* = 505 Adolescents aged 15–21 years Mean = 18.6 years Low income and from minorities in New York City United States	Distribution of pBMI, *n* (%) Underweight: 54 (10.7) Normal: 265 (52.5) Overweight: 96 (19) Obesity: 90 (17.8) IOM 2009	Excessive GWG in 50% of participants IOM 2009	pBMI was associated positively with GWG Association between pBMI and excessive GWG.	Good, 9
Danilack et al. [33]. 2018 Retrospective cohort, 2007–2008	*n* = 91 Adolescents aged ≤17 years Mean = 16.5 years, Different ethnicities United States	Distribution of pBMI, *n* (%) Low weight: 3 (3.3) Normal: 55 (60.4) Overweight: 22 (24.4) Obesity; 11 (12.1) CDC 2007	GWG in kg: Mean: 15.5 ± 6.3 Low weight: 13 Normal: 15 Overweight: 17 Obesity: 12 (*p* = 0.171).	pBMI was not associated with GWG There was no linear trend in the averages of GWG kg per pBMI.	Good, 11
Ekambaram et al. [35]. 2018 Retrospective cohort, 2012–2013	*n* = 411 Adolescents aged 13–18 years Low-income urban residents, 90% Afro-American United States	Distribution of pBMI in % Underweight 5.6 Normal: 58.2 Overweight: 21.1 Obesity: 15.1 WHO.	GWG in % Excessive: 53 Adequate: 29 Below: 17.8 GWG: 15.6 ± 6 kg. GWG general (%) Inadequate: 17.8 Adequate: 29.3 Excessive: 53 Excessive GWG according to pBMI Underweight: 33 Normal: 42.6 Overweight: 67.9 Obesity: 77.8, *p* < 0.001 IOM 2009	pBMI was associated positively when analyzed by categories, but not with GWG when analyzed overall.	High, 13
Joseph et al. [36]. 2008 Retrospective cohort, 2002–2005	*n* = 102 Adolescents aged 15–21 years; average, 15 years Vulnerable urban zones, different ethnicities United States	Distribution of pBMI in % Low weight: 19 (20%) Normal: 52 (54%) Overweight: 18 (18.6%) Obesity: 8 (8.2%) IOM 1990	Total GWG in % Excessive: 36 Adequate: 30 Below: 34 Excessive GWG according to pBMI (%) Underweight: 26.5 Normal: 36.5 Overweight: 66.5 Obesity: 25 Inadequate or low GWG according to pBMI (%) Underweight: 31.6 Normal: 34.6 Overweight: 16 Obesity: 25 IOM 1990	pBMI was associated positively by with GWG categories, particularly among those who were overweight.	High, 13
Chu et al. [43]. 2009 Retrospective cohort, 2004–2005	*n* = 5861 Adolescents aged 14–19 years A representative sample of pregnant women United States	Distribution of pBMI, in % Underweight: 9.9 ± 0.6 Normal: 59.3 ± 1.0 Overweight: 19 ± 0.8 Obese: 11.7 ± 0.7 NHLBI 1998	GWG in pounds in the adolescent group Mean GWG: 16.1 Those with an overweight pBMI had a higher frequency (60%) of excessive GWG than those of normal 40%	pBMI was associated positively by with GWG categories, particularlly among those overweight, regardless of their chronological age.	High, 12
Groth et al. [38]. 2017 Prospective cohort, 2009–2014	*n* = 360 Adolescents aged ≤20 years Low income, African American, and primiparous United States	Distribution of pBMI, (%) Underweight: 1.7 Healthy weight: 75.5 Overweight: 15.3 Obesity: 7.5% CDC 2007	GWG in % Excessive: 38 Adequate: 29 Below: 33 GWG in kg: 13.7 ± 7 GWG in kg by pBMI Underweight: 13.5 Normal: 13.8 Overweight: 14 Obesity: 13 IOM 2009	pBMI was not associated with absolute GWG.	Good, 9
Sámano et al. [40]. 2017 Prospective cohort, 2009–2016	*n* = 168 Adolescents aged 12–17 years Low income, without public medical services, government support, or health insurance Mexico	Distribution of pBMI, in % Underweight: 4 Normal: 75 Overweight: 17 Obesity: 4 IOM 2009	Distribution of GWG Median of GWG (kg) Underweight: 13 Normal: 12 Overweight: 11 Obesity: 4 Excessive GWG (%): Underweight: 0 Normal: 25 Overweight: 42 Obesity: 33 Insufficient GWG, Underweight: 0 Normal: 35 Overweight: 25 Obesity: 67 IOM 2009	pBMI was associated positively with GWG by categories.	High, 14
Maternal age					
Elchert et al. [9]. 2015 Retrospective cohort, 2006–2012	*n* = 72,126 Pregnant women stratified by maternal age: *n* = 979: <15 years *n* = 22,845: 15–17 years *n* = 48,302: 18–19 years Representative sample United States	Distribution of pBMI in % <15 years old: 23.2 15–17 years old: 23.5 18–19 years old: 24.5 WHO age- and sex-specific BMI	GWG in % Excessive GWG <15 years: 59.8 15–17 years: 59.9 18–19 years: 62.6 GWG in kg <15 years: 14.9 15–17 years: 15.8 18–19 years: 16.3 Definition IOM 2009	Risk (aOR) of low GWG <15 years: 1.12 (95% CI: 1.01–1.51) 15 to 17 years: 1.33 (95% CI: 1.27–1.40) 18 to 19 years: 1.26 (95% CI: 1.21–1.30). All compared with adult mothers, *p* < 0.001	High, 12
Groth et al. [34]. 2008 Retrospective cohort, 1990	*n* = 330 Adolescents aged 12–19 years. African Americans, low-income, primiparous United States	Age in three categories: <16 years *n* = 106, 16–17 years, *n* = 146; 18–19 years, *n* = 78	GWG in kg by age <16 years: 13.7 16–17 years: 14.1 18–19 years: 13.8 There were no differences in the mean GWG (kg) by age	Chronological age was not associated with GWG.	High 12
Number of deliveries (parity)					
Chu et al. [43] 2009 Retrospective cohort, 2004–2005	*n* = 5861 Adolescents aged 14–19 years Representative sample United States	Three groups for the number of pregnancies: Primiparous (0), 1–2 deliveries, ≥3 deliveries	GWG in kg Excessive GWG % Primiparous: 20.1 1–2 births: 12.7 ≥3 births: 10.8 (*p* < 0.001). Below GWG % Primiparous: 23.2 1–2 deliveries: 16.8 ≥3 deliveries: 11.5 (*p* < 0.001)	Number of deliveries was associated with GWG categories. Association between excessive GWG and parity: 1–2 births β = −3.15, SE = 0.20 *p* < 0.001; ≥3 births β = −4.27, SE = 0.35, *p* < 0.001. Excessive GWG was considered if above pBMI Normal gain: >15.75 Overweight gain: >11.25 kg	High, 12
Timur et al. [37]. 2016 Retrospective cohort, 2010–2014	*n* = 66 Adolescents aged 16–19 years Turkey	Maternal BMI (kg/m^2^) by parity: second pregnancy, 25 (20–37)/ first pregnancy, 23 (19–35) *p* = 0.672	GWG (kg) showed no difference between the second pregnancy (11.5 ± 5.8) and first pregnancy (12.4 ± 5.2), *p* = 0.462	First and second pregnancy and GWG were not associated.	Good, 11
Diet					
Sámano et al. [40]. 2017 Prospective cohort, 2009–2016	*n* = 168 Adolescents aged 12–17 years Low income, without public medical services, government support, or health insurance Mexico	Adequacy of energy, as a percentage	Distribution of GWG Median of GWG (kg) Underweight: 13 Normal: 12 Overweight: 11 Obesity: 4 Excessive GWG (%): Underweight: 0 Normal: 25 Overweight: 42 Obesity: 33 Insufficient GWG, Underweight: 0 Normal: 35 Overweight: 25 Obesity: 67 IOM 2009	Percentage of energy adequacy was not associated with GWG. The percentage of energy adequacy, serum leptin, and pregestational weight explained the GWG. R^2^ = 0.192, SE = 3.99 (95% CI 14.89–30.890, *p* = 0.001) for the difference between pre-pregnancy and maximum gestational weight in kg.	High, 14
Variables related to cardio-metabolic risk					
Noreña et al. [39]. 2018 Prospective cohort, 2009–2016	*n* = 40 Adolescents aged 14–17 years Primiparous, low income Colombia	Leptin, insulin, and adiponectin in the second trimester determined by ELISA (ng/mL)	Leptin, insulin, and HOMA-IR were associated with pBMI, but not with GWG	A positive correlation (*p* < 0.001) was found between leptin levels and pBMI (r = 0.839). A positive correlation was observed between pBMI and insulin levels (r = 0.56; *p* ≤ 0.001), and between the HOMA-IR index and pBMI (r = 0.54; *p* = 0.0003).	Regular, 7
Sámano et al. [40]. 2017 Prospective cohort, 2009–2016	*n* = 168 Adolescents aged 12–17 years Low income, without public medical services, government support, or health insurance Mexico	Serum leptin in the last trimester determined by ELISA (ng/mL)	Distribution of GWG Median of GWG (kg) Underweight: 13 Normal: 12 Overweight: 11 Obesity: 4 Excessive GWG (%): Underweight: 0 Normal: 25 Overweight: 42 Obesity: 33 Insufficient GWG, Underweight: 0 Normal: 35 Overweight: 25 Obesity: 67 IOM 2009	Leptin from the last trimester was associated positively with GWG. Higher leptin concentrations in the last trimester of gestation were associated with higher GWG (R^2^ = 0.177, *p* < 0.001) Leptin explained GWG SE = 0.03 (95% CI 0.100–0.248). GWG (%) was determined by the difference between maximum gestational and pre-pregnancy weight in kg. IOM 2009.	High, 14

**Table 2 nutrients-15-01530-t002:** Synthesis of studies on family and sociocultural characteristics and gestational weight gain in pregnant adolescents.

Author, Year, Study Design	Participants	Exposure	GWG	Findings	Quality
Sam-Soto et al. [41]. 2015 Cross-sectional, 2010–2013	*n* = 165 Adolescents aged 10 to 16 years With and without a history of sexual abuse. from vulnerable urban zones Mexico	Adolescents with or without a history of sexual abuse Adolescents who lived or did not live with their father	Mean GWG (kg) Sexually abused: 7.5 Non-abused adolescents: 12.5, *p* = 0.005.	History of sexual abuse was related to low GWG. The adolescents who were most likely to have been sexually abused had lower socioeconomic status and did not live with their father.	Good, 11
Sámano et al. [42]. 2019. Cross-sectional, 2008–2014	*n* = 352 Adolescents aged 12 to 18 years From low resources, without public medical services, government support, or health insurance Mexico	Family support network, divided in quartiles (Q) according to the support network size Energy intake (kcal)	GWG (%) by IOM 2009 Excessive: 23.5 Adequate: 38 Below: 37.5 GWG (%) by IOM 2009 in quartile (*p* = 0.003) Q I: Excessive: 26 Adequate: 43 Below: 34 Q II: Excessive: 30 Adequate: 37 Below: 33, Q III: Excessive: 20 Adequate: 34 Below: 46 Q IV: Excessive: 25 Adequate: 38 Below: 37	Family support by quartile was associated with GWG but showed a non-linear trend. The type of members in each quartile was uncertain.	High, 13
Danilack et al. [33]. 2018 Retrospective cohort, 2007–2008	*n* = 91 Adolescents aged 17 years, mean age: 16.5 years United States	Race/ethnicity (%) Latina: 55 African American: 18.7 White: 14.3 Other: 12.1	GWG kg: 15.5 ± 6.3 Low weight: 13 Normal: 15 Overweight: 17 Obesity: 12 (*p* = 0.171)	There was no average GWG (in kg) per maternal race/ethnicity. BMI was similar in all racial/ethnic groups.	Good, 11
Cunningham et al. [32]. 2018 Retrospective cohort, 2008–2012.	*n* = 505 Adolescents aged 15–21 years, mean age: 18.62 years. Low income and minorities in New York City United States	Race/ethnicity (%) Latina: 266 (52.7%) Black, non-Latina: 196 (38.8%) Other: 43 (8.5%)	Excessive GWG was present in *n* = 255 (50%) By race/ethnicity Latina: 135 (52.9%) Black-non-Latina: 100 (39.2%) Other: 20 (7.8%) Overweight: β 2.41, SE 1.06 *p* < 0.05 Obese: β 2.58, SE 1.08 *p* < 0.05 IOM 2009	Race/ethnicity was associated with excessive GWG (Latina group), maybe due to pBMI.	Good, 9
Joseph et al. [36]. 2008. Retrospective cohort, 2002–2005.	*n* = 102 Adolescents aged 15–21 years, mean age: 15 years. Vulnerable urban zones United States	Race/ethnicity (%) African-American: 84 Latina: 12 Non-Hispanic white: 1% Other ethnicities: 3%	GWG in % Excessive: 36 Adequate: 30 Below: 34 IOM 1990	There was no average GWG (in kg) per maternal race/ethnicity.	High, 13

## Data Availability

All data generated or analyzed during this study are included in this published article and its Appendix A.

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
