# Peer review of "Individual, Family, and Social Factors Associated with Gestational Weight Gain in Adolescents: A Scoping Review"

_nutrients, 2023, doi:10.3390/nu15061530_

Round 1

Reviewer 1 Report

This is an interesting study on a hot topic with multiple long-term consequences, for both adolescent mothers and their offspring. However, there are some issues that need addressing:

-          The manuscript needs major proofing from a proficient English user

-          Row 20 – replace ”inappropriate” with ”inadequate”

-          Row 50 – replace ”level” with ”burden”

-          Row 60 – replace ”remain” with ”persist”

-          Row 63 – please use reference when mentioning the Institute of Medicine – what kind of institute is this?

-          Row 81 – please replace ”such an interesting event” with a more appropriate turn of words

-          Rows 90-92 – please review and rephrase

-          Row 97 – replace ”rest” with ”remaining”

-          Row 97 – replace ”to their data’s analysis”, it doesn’t make sense

-          Figure 1 – move the title of the figure underneath. The top box on the right says ”Excluded cases”, but no reason is given - please add motivation. The bottom box on the right needs to be larger to incorporate the whole text (either that, or make the font smaller)

-          Row 102 – please rephrase: ”The following databases were consulted”

-          Row 121 – replace ”higher” with ”highest”

-          Rows 129-130 – most of the text is a repeat of the Methods – please remove

-          Row 136 – the authors claim to have included a ”representative” sample of adult women -why?

-          Row 141 – remove ”made”

-          Row 142 – remove reference 32 from the study made in Columbia – it is a duplicate of the Turkish study referenced before

-          Row 152 – replace ”identify” with ”identified”

-          Row 153 – replace ”had” with ”having”

-          Rows 158 and 281 – replace ”Afro” with ”African”

-          Row 171 – replace ”mood” with ”psychological”

-          Rows 193-194 – please rephrase

-          Row 228 – replace ”caution” with ”cautious”

-          Row 237 – replace ”a” with ”one” and include ”lower parity [equals] higher GWG”

-          Rows 241-246 – please rephrase

-          Row 251 – replace “find” with “found”

-          Rows 262-264 – please rephrase

-          Row 268 – replace “vias” with “pathways”

-          Rows 292-293 – please rephrase

-          In section 4.3 the issue of concealment of pregnancy and the suboptimal weight gain in this case should be addressed. Also, I find that the discussion about food cravings is more of a cultural issue, rather than social, and it doesn’t belong here

-          Row 323 – replace “it” with “there”

-          Rows 332 and 340 – rephrase points 2) and 3) as recommendations, then explain the rationale behind it

-          Row 376 – replace “obesity” with “obese”.

Author Response

Responses to reviewer

We thank the reviewers for their time reviewing this manuscript. We regret that our revisions are not marked up with the Track Changes function because the revised manuscript was first sent to the English language editor, so the marked-up changes disappeared. Instead, in each suggestion, we added the number of the line in the text where you can find the revisions.

Reviewer 1.

Row 20 – replace “inappropriate” with “inadequate”

Reply: The replacement was made. Besides, we added that the "inappropriate" term refers to "insufficient" and "excessive" gestational weight gain.

Row 50 – replace “level” with “burden”

Reply: Accepted

Row 60 – replace “remain” with “persist”

Reply: Accepted

Row 63 – please use reference when mentioning the Institute of Medicine – what kind of institute is this?

Reply: Accepted. Lines 68 to 73

Row 81 – please replace “such an interesting event” with a more appropriate turn of words

Reply: Accepted. Lines 92 to 93

Rows 90-92 – please review and rephrase

Reply: The paragraph has already been reformulated. Lines 98 to 103

Row 97 – replace “rest” with “remaining”

Reply: The replacement has already been done. Line 117

Row 97 – replace “to their data’s analysis”, it doesn’t make sense

Reply: The replacement has already been done. Lines 118 to 119

Figure one – move the title of the figure underneath. The top box on the right says, “Excluded cases”, but no reason is given - please add motivation. The bottom box on the right needs to be larger to incorporate the whole text (either that, or make the font smaller)

Reply: The proposed changes to Figure 1 have already been made.

Row 102 – please rephrase: “The following databases were consulted”

Reply: The proposed change has already been made. Line 110

Row 121 – replace “higher” with “highest”

Reply: The proposed change has already been made.

Rows 129-130 – most of the text is a repeat of the Methods – please remove

Reply: The proposed deletion was carried out

Row 136 – The authors claim to have included a “representative” sample of adult women - why?

Reply: The manuscript specifies that the sample of women belongs to a national survey, so it was considered representative. Lines 145 to 146.

Row 141 – remove “made”

Reply: Deleted already done

Row 142 – remove reference 32 from the study made in Columbia – it is a duplicate of the Turkish study referenced before

Reply: The proposed correction was made

Row 152 – replace “identify” with “identified”

Reply: Replacement was made

Row 153 – replace “had” with “having”

Reply: Replacement was made

Rows 158 and 281 – replace “Afro” with “African”

Reply: Replacement was made. Lines 166 to 168, 262 and 314

Row 171 – replace “mood” with “psychological”

Reply: Replacement was made.

Rows 193-194 – please rephrase

Reply: Observation accepted

Row 228 – replace “caution” with “cautious”

Reply: Replacement was made

Row 237 – replace “a” with “one” and include “lower parity [equals] higher GWG”

Reply: Replacement was made. Lines 328 to 329

Rows 241-246 – please rephrase

Reply: Observation accepted

Row 251 – replace “find” with “found”

Reply: Replacement was made

Rows 262-264 – please rephrase

Reply: Observation accepted

Row 268 – replace “vias” with “pathways”

Reply: Replacement was made

Rows 292-293 – please rephrase

Reply: Observation accepted

In section, 4.3 the issue of concealment of pregnancy and the suboptimal weight gain, in this case, should be addressed. Also, I find that the discussion about food cravings is more of a cultural issue, rather than a social one, and it does not belong here.

Reply: The observation proposed in section 4.3 was made

Row 323 – replace “it” with “there”

Reply: Replacement was made

Rows 332 and 340 – rephrase points 2) and 3) as recommendations, then explain the rationale behind them.

Reply: Observation made

Row 376 – replace “obesity” with “obese”

Reply: We changed the sentence, it now reads “…especially if adolescents began their first pregnancy with overweight or obesity…”

Sincerely

Professor Hugo Martinez Rojano

Reviewer 2 Report

The topic of the study is quite interesting and important, taking into account the presented problem of improper weight gain during pregnancy in adolescents. In my opinion, both too high increase and too low increase should be considered. I am not sure if the terms used for finding articles and further examination was properly done. However, after proper rewriting it could be resubmitted. Some of additional comments are presented below.

1. Abstract is too long and should be rewritten.

2. Introduction is too short for me, it should better describe the problem.

3. Lines 57-59, 83-84: The sentences should be corrected. Generally, English in the manuscript is really poor, sometimes difficult to understand. Whole manuscript should be corrected by an English professional.

4. Line 63: The IOM should be better explained (see Abstract).

5. Lin 72: The classification of GWG should be described in details.

6. What is the meaning of the word "parity" in your work?

7. Results and Discussion are difficult to follow. They should be carefully rewritten.

8. Conclusions are really poor. I am afraid the results are not properly discussed and thus, it is difficult to have the proper conclusions.

9. Table 1 should be more comprehensive.

10. Maybe some schemes/ graphs could help to better understand the data presented.

Author Response

Reviewer 2

We thank the reviewers for their time reviewing this manuscript. We regret that our revisions are not marked up with the Track Changes function because the revised manuscript was first sent to the English language editor, so the marked-up changes disappeared. Instead, in each suggestion, we added the number of the line in the text where you can find the revisions.

  1. The abstract is too long and should be rewritten.

Reply: The abstract was rewritten according to the reviewer's proposal. Lines 20 to 34

  1. The introduction is too short for me; it should better describe the problem.

Reply: The introduction was rewritten according to the reviewer's proposal. Lines 43 to 81

  1. Lines 57-59, 83-84: The sentences should be corrected. Generally, the English in the manuscript is really poor, and sometimes difficult to understand. The whole manuscript should be corrected by an English professional.

Reply: The sentences were corrected according to the proposal.

  1. Line 63: The IOM should be better explained (see Abstract).

Reply: The manuscript explains that it is the Institute of Medicine of the United States of America. Lines 68 to 73.

  1. Lin 72: The classification of GWG should be described in detail.

Reply: The classification of gestational weight gain has already been described. Lines 93 to 96

  1. What is the meaning of the word "parity" in your work?

Reply: The meaning of parity was explained in the manuscript. It is the number of deliveries

  1. Results and Discussion are difficult to follow. They should be carefully rewritten.

Reply: We agree. Now, we present a manuscript previously reviewed and corrected by a proofreader whose English was his native language, as suggested by one reviewer.

  1. The conclusions are really poor. I am afraid the results are not properly discussed and thus, it is difficult to have the proper conclusions.

Reply: The results were discussed in greater depth so that the conclusions are clearer.

  1. Table 1 should be more comprehensive.

Reply: Changes were made to table 1 to make it more understandable.

  1. Maybe some schemes/ graphs could help to better understand the data presented.

Reply: Figure 2 was included to make the manuscript more understandable.

Sincerely

Professor Hugo Martinez Rojano

Round 2

Reviewer 2 Report

Dear Authors,

You have revised your manuscript thoroughly and now it sounds much better. Now it is easy to follow your ideas and the topic seems very interesting. I am still not convinced that the Conclusions are properly prepared. The last subchapter of Discussion seems to me to contain the correct conclusions of the article. Perhaps you can modify Conclusions according to this Strength and Limitations section. Generally, Conclusions should be one or two paragraphs long, not as many as now.

lines 167, 258, 261 - please add the citation directly after "et al." (e.g., Hill et al. [55])

Author Response

Responses to Reviewer

We appreciate all your proposals and the time invested in revising our manuscript. The suggestions are marked with tracked changes in the manuscript, and we add the number of the line in the text where you can find the changes.

Reviewer 2

I am still not convinced that the conclusions are properly prepared. It seems to me that the last discussion subchapter contains the correct conclusions of the article. You may be able to modify the conclusions based on this strengths and limitations section. Generally, the conclusions should have one or two paragraphs, not as many as now.

Reply: The conclusions were rewritten according to your suggestion.

Lines 167, 258, 261 - add the citation directly after "et al.," (e.g. Hill et al. [55]).

Reply: The citation was added directly after et al. (lines 171, 261, 265), as well as in the tables.

Sincerely

Professor Martínez-Rojano H.
